# Electronic Devices for Stress Detection in Academic Contexts during Confinement Because of the COVID-19 Pandemic

Cristhian Manuel Durán-Acevedo * 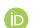, Jeniffer Katerine Carrillo-Gómez and Camilo Andrés Albarracín-Rojas

Multisensor System and Pattern Recognition Research Group (GISM), Electronic Engineering Program, Universidad de Pamplona, Km 1 Vía Bucaramanga, Pamplona 543050, Colombia; jeniffer.carrillo@unipamplona.edu.co (J.K.C.-G.); camilo.albarracin@unipamplona.edu.co (C.A.A.-R.)
* Correspondence: cmduran@unipamplona.edu.co; Tel.: +57-3112135846

**Abstract:** This article studies the development and implementation of different electronic devices for measuring signals during stress situations, specifically in academic contexts in a student group of the Engineering Department at the University of Pamplona (Colombia). For the research's development, devices for measuring physiological signals were used through a Galvanic Skin Response (GSR), the electrical response of the heart by using an electrocardiogram (ECG), the electrical activity produced by the upper trapezius muscle (EMG), and the development of an electronic nose system (E-nose) as a pilot study for the detection and identification of the Volatile Organic Compounds profiles emitted by the skin. The data gathering was taken during an online test (during the COVID-19 Pandemic), in which the aim was to measure the student's stress state and then during the relaxation state after the exam period. Two algorithms were used for the data process, such as Linear Discriminant Analysis and Support Vector Machine through the Python software for the classification and differentiation of the assessment, achieving 100% of classification through GSR, 90% with the E-nose system proposed, 90% with the EMG system, and 88% success by using ECG, respectively.

**Keywords:** stress; academic contexts; GSR; ECG; EMG; E-nose; pattern recognition; SVM

## 1. Introduction

Stress is considered as a physiological reaction in which different defense mechanisms interact during the confrontation of a situation or imminent threat that stimulates a fight or escape response of the body [1]. Lazarus and Folkman [2] define it as "A particular reaction between a person and his surroundings considered as threatening or that overpass the available resources and jeopardize his wellness". The academic stress comes out toward typical problems in the educational context, this can be a necessary and natural reaction for fulfilling the demands required and in which factors such as the academic overcharge, group projects, competitivity, lack of economic resources, and the deficient time organization take part [3]. Stress levels can increase in significant proportions in some students, especially during the exam period [4].

Additionally, academic stress is associated with different negative results on the person, including deficiencies in academic performance, the daily homework, as well as detriment in physical and mental health when the person is often involved in stressful situations [5,6]. Several types of research have been focused on the health changes related to academic stress, discovering results that suggest that the physical response to stress plays an important role in health and these must be considered in the design of Promotional Health Programs [7,8]. Mental health professionals think stress may help people to solve some problems, however, it makes people irritable, upset, and, in serious cases, stress keeps a close relation with diseases such as diabetes, depression, sleep disorders, heart illnesses, and gastric problems [9–11].

The quick spread of COVID-19 forced the closure of schools and universities all over the world, driving to a virtual form in the teaching method, continuing the educational

process from home. In Colombia, according to United Nations Educational, Scientific and Cultural Organization (UNESCO), near 2 million students continue their formative process in virtual modality [12]. The mobility restriction imposed in quarantine, along with the challenges generated by the transition from a face-to-face education system to a virtual one, can be stressing factors in addition to the prolonged confinement, the insufficient explanation by the teacher, and lack of materials and tools such as a computer, internet, among others. Not to mention, every student is responsible for his educational process [4,13,14].

The early detection of mental stress can prevent health problems, helping patients to become skilled in the management and confrontation of situations or events perceived as stressful. Besides, it provides an important advance in the quality of life, giving an emotional response assertively toward the daily situations with the assistance and supervision of qualified staff [15,16].

Psychometrics questionnaires have been used traditionally as instruments for measuring mental stress, these are useful for understanding the individual differences in stress perception, as well as the stimulus frequency that can be perceived as stressing [17,18]. The use of the internet for providing psychometric questionnaires gives several advantages for the evaluation and potential monitoring of the research survey respondents, avoiding, in this way, physical contact. Besides, they can be applied confidently and accurately in a digital format due to their equivalence with the traditional paper version questionnaires [19].

However, psychometric instruments give subjective solutions that may delay the treatment, since these can be evaluated wrongly due to the participant's unwillingness for resolving the questionnaire, so, it is important to have objective methods for quantifying the stress level [20].

When the person perceives a stressful situation, the main physiological functions of the body get stimulated, such as blood pressure and breathing, among others. These functions are regulated by the autonomic nervous system (ANS). One of its branches, the sympathetic nervous system (SNS), moves the body's resources, preparing it for giving an opportune response under stressful conditions [21].

The physiological parameters affected by the SNS have been used in a lot of studies for stress detection, among which are: Salivary Cortisol [22], respiratory signs [23,24], skin temperature and thermal image [25–27], variability of heart rate (HRV) [28–30], electrocardiogram (ECG) [15,31,32], electroencephalogram (EEG) [20,33,34], Galvanic Skin Response (GSR) [35–37], pupil diameter [38,39], electromyogram (EMG) [40–42], and the Volatile Organic Compounds (VOC's) emitted by the skin [43–45]. The response focus of each one of these physiological signals depends on the context since these signals can show variations due to physical activity. For example, a participant can show a higher heart rate when standing than when sitting. This signal also increases when the participant perceives a mentally stressing situation, that is why it is important to define the context and the situation in which the physiological measurement will be taken [21,46].

Perspiration is mainly associated with the human body's thermoregulation; in the emotional sweat, the mental efforts stimulate the production of acetylcholine, which is a neurotransmitter that stimulates the secretion of abundant water in the endocrine glands. Sweat gives information about several biomarkers biologically relevant that may serve as indicators or disorders [47]. When sweat is induced by an emotional stimulus such as stress, the behavior of sweat glands and the assessment of the sweat rate are useful to estimate the mental stress level in human beings [48].

The cortisol in the biofluids and the VOC's emitted by skin seem to be useful markers for the detection of emotional stress. The VOC's detection technologies have been used widely in the medical field for the different disorders and diseases detection that show changes in the sweat VOC's pattern [43].

The forehead has been studied as a possible measuring location for registering changes in the VOC's associated with psychological stress. There have been VOC's samples ana-

lyzed with conventional techniques such as thermal desorption gas chromatography in two layers and mass spectrometry, determining that VOC's profiles change with stress [44]. One of the few studies done with the implementation of an electronic nose was conducted by Cortes, in which the use of an electronic nose is described for the detection of stress bio-markers such as adrenaline and cortisol [49].

Among the most used physiological signals in the research for the detection of stress levels, there is heart activity, since it varies according to the intensity of the stimulus perceived in response to the changes in the autonomic nervous system (ANS). For heart activity measuring, the ECG is used through electrodes placed on the body [25].

For the extraction of the ECG signal features, the R peak is used, among which there are the Heart Rate, RR Interbeat interval (IBI), and the heart rate variability (HRV). The HRV is the variation in time between consecutive beats sequences (RR intervals). In healthy people, it varies continuously, but when the ANS activity is altered, the HRV values decrease [16,50].

Through a stressful situation, the electrical activity can be increased in specific muscles in comparison to a no-stress situation [41]. Different researchers have suggested the EMG trapezius muscle activity as a stress physiological marker, given that the stressing events induce involuntary reactions in facial and trapezius muscles [51]. Other researchers indicate that EMG signals give more relevant information in comparison to respiratory signs for determining the stress level [52].

This article presents an investigation about the development of an Electronic Nose (E-nose) for stress detection in university students, using emotional sweat as a base. The change in the VOC's profile is emitted by the skin, specifically on the forehead during the perspiration process [44,48]. On the other hand, a Grove 1.2 GSR sensor (Seeed Technology Co., Ltd., Shenzhen, China) was implemented from the "Speed studio" for the galvanic skin response in measuring fingertips. The GSR sensor response is based on the electrical activity in the skin that is also affected by emotional sweat [53]. Furthermore, an electromyography surface device with amplification and filtration stages was designed for measuring the electrical response of the trapezius muscle, placing the electrodes according to the positions set in the Surface ElectroMyoGraphy for the Non-Invasive Assessment of Muscles (SENIAM) regulation, in which also the ADS1298ECG-FE (Texas Instruments, Dallas, TX, USA) from the "Texas Instruments" company was implemented for measuring and gathering the electrocardiographic signal to identify the heart rate variability (HRV). For the GSR and EMG data gathering, a graphic interface was designed where gathered data can be visualized and the pathway in which they were stored can be chosen. These four methods: E-nose, GSR, EMG, and HRV, are proposed to acquire a larger quantity of each participant's information, and therefore, every response acquired in the electronic devices is validated.

Furthermore, for the data processing, the free software Python 3.8 version was used, which was implemented and developed some pattern recognition and artificial intelligence algorithms [54]. As a validation technique, the SISCO inventory of academic stress was implemented. This inventory has been subjected to different validation and reliability measures, as well as the statistical and psychometric analyses that endorse its applicability in the academic ambit [55–57].

## 2. Materials and Methods

### 2.1. Measurement Protocol and Volunteers Selection

For this research, 25 students from the engineering and architecture faculty of the University of Pamplona participated as volunteers, 7 women and 18 men respectively, aged between 18 and 30 years (see Table 1). Regarding the inclusion criteria, each student must have good health, be over 18 years and they should have virtual classes to take the exam. On the other hand, about exclusion criteria, the students must not be smokers, and they should not use psychoactive substances nor suffer from any psychological disorder. At the recruitment stage, they were instructed to abstain from consuming alcoholic or carbonated

beverages, and medicines. Additionally, they were told not to use perfumes, sunblock, lotion and they should sleeping adequately.

**Table 1.** Information about the participants.

| No. | Label | Age | Gender |
| --- | --- | --- | --- |
| 1 | A | 22 | Male |
| 2 | B | 25 | Male |
| 3 | C | 19 | Female |
| 4 | D | 18 | Male |
| 5 | E | 25 | Male |
| 6 | F | 24 | Female |
| 7 | G | 23 | Male |
| 8 | H | 21 | Male |
| 9 | I | 20 | Male |
| 10 | J | 24 | Male |
| 11 | K | 27 | Male |
| 12 | L | 21 | Female |
| 13 | M | 26 | Male |
| 14 | N | 24 | Male |
| 15 | O | 23 | Male |
| 16 | P | 24 | Male |
| 17 | Q | 23 | Male |
| 18 | R | 23 | Female |
| 19 | S | 30 | Female |
| 20 | T | 24 | Female |
| 21 | U | 23 | Male |
| 22 | V | 23 | Male |
| 23 | W | 24 | Male |
| 24 | X | 20 | Female |
| 25 | Y | 22 | Male |

Finally, we obtained informed consent by hand from the participants for the data acquisition and processing measures for the stress state during a virtual exam performance. For the relaxation state, some measurements were taken during the academic semester between March and June of 2020. The physiological measurements were acquired in the student's homes, following the necessary security actions like the use of overalls, masks, and disposable gloves.

It is necessary to mention that experiment and evaluation of the electronic devices developed in this research were proposed as a pilot study during the COVID-19 pandemic, therefore, the number of samples was not established or estimated as a target since the participation was limited to the availability of the students who were located in their current places of residence or elsewhere in the Pamplona city, making the sample collection much more efficient.

### 2.2. Galvanic Skin Response (GSR)

Skin conductance is one of the most used methods in psychophysiological research, it is also called skin electrical activity, and refers to each of the skin electrical properties as a response to the sweat secretion by sweat glands. Eccrine glands are mainly stimulated in response to emotional events such as stress, these glands are distributed over all of the body in low densities, with a larger concentration in the face, in palms of hands, soles of feet, and armpits, with the palms of hands being the preferred location for the GSR measurement [53]. Due to the existence of electrolytes in sweat, the electric resistance decreases, and the skin conductance increases—this response is directly associated with the sympathetic nervous system (SNS) that responds during emotionally stressful stimulus [58].

For taking the measurements, electrodes attached to a device were used, locating them on the index and middle finger on the participant's non-dominant hand (see Figure 1).

Throughout the tests, there was a five-minute time-lapse estimated for measuring, with signals acquired over a long time in a relaxation state where the sensor is set, keeping constant the shape of the wave without presenting novelties as long as the participant stays calm. The measures were acquired at an around 10 samples per second sampling rate for 5 min, the participant was told to avoid touching the electrode fastener and not to make movements with those fingers.

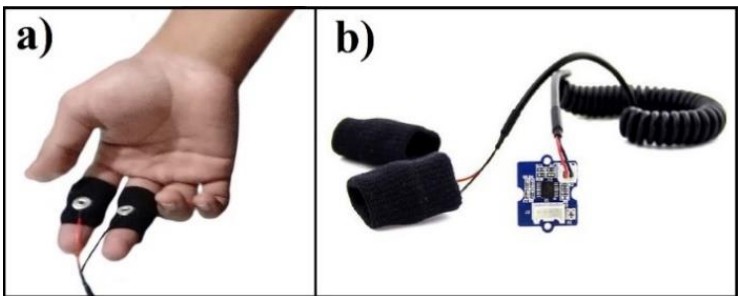

**Figure 1.** Galvanic Skin Response (GSR) Sensor: (**a**) location of the electrodes; (**b**) electronic card and electrodes.

### 2.3. Electronic Nose System (E-Nose)

An electronic nose system is based on the gas sensor's implementation with different sensibility levels, located in a measuring camera; commonly, the electronic system is combined with pattern recognition algorithms and artificial intelligence to find a characteristic profile that allows classifying VOCs. Usually, sensors based on metal oxide semiconductors are among the most used due to the wide variety of compounds that it can detect and to its wide commercial diversification [59].

### 2.3.1. Design

Figure 2 shows the measuring electronic circuit for the gas sensors. For the generated signal in each sensor, an Arduino Astar 32U4 card (Pololu, Las Vegas, NV, USA) made by "Pololu" company was connected to the circuit, it has 8 analog inputs with 10 bits resolution. All used sensors share the same circuit configuration given by the manufacturer; however, the power consumption of every sensor is different according to their electric consumption since the sensor's matrix feeds from a direct voltage source of 5 V with 2A capacity. The sensor's voltage response is measured in the load resistance RL = 1 kΩ.

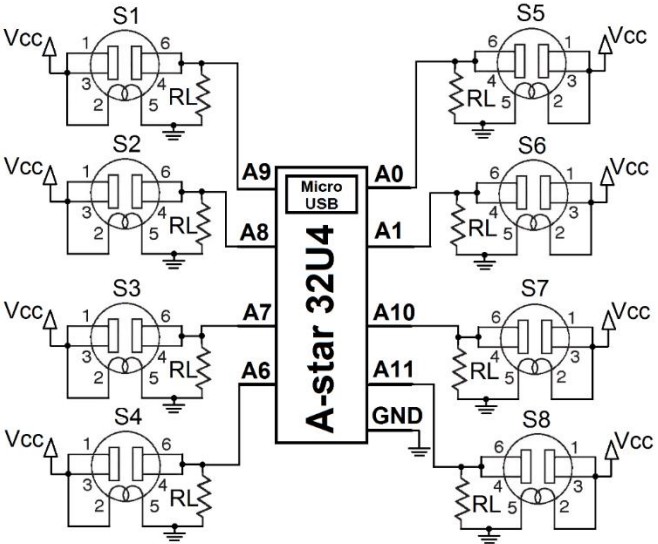

**Figure 2.** Measurement circuit for gas sensors.

Metal Oxide (MOX) gas sensors manufactured by Hanwei and Figaro companies were used, where each sensor's information is shown in Table 2. The E-nose consists of a measuring chamber that comprises a sensor array that can detect the organic compounds which are controlled by pneumatic valves. The measuring chamber was made of stainless-steel, which was connected to a vacuum pump with an independent power supply of 6 Volts Direct Current (VDC) and 500 mAh feedback, controlled from the Arduino device with a Bipolar Junction Transistor (BJT) (Ref: TIP41) transistor configured as a switch.

**Table 2.** Chemical-resistive sensors for the Electronic Nose (E-nose).

| Label | Sensor Reference | Specific Targets |
|---|---|---|
| S1 | MQ2 | Propane, Methane, Alcohol, Hydrogen |
| S2 | MQ3 | Alcohol, Benzine, CO, CH4 |
| S3 | MQ4 | Methane, Natural Gas |
| S4 | MQ5 | Natural Gas, GLP |
| S5 | MQ9 | CO, Flammable gas |
| S6 | MQ138 | Toluene, Acetone, Ethanol, and Formaldehyde |
| S7 | TGS825 | Hydrogen sulfide |
| S8 | TGS832 | Chlorofluorocarbons |

Figure 3 shows the vacuum pump feedback circuit to connect it with the Arduino card for supplying the necessary current. The VOC's were carried out to the measuring chamber through a piping circuit and using a vacuum pump.

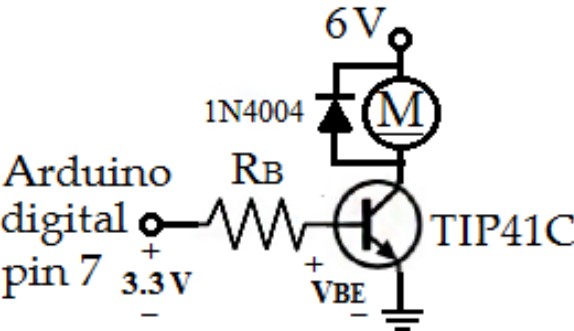

**Figure 3.** Trigger circuit for vacuum pump.

The system controls the vacuum pump with the aim of limiting the Arduino output current to less than 15 mA, whereas the base resistance $R_B$ was calculated with Equation (1). The above allows to extend the device's useful life and guarantee the transistor to keep in optimal working conditions.

$$I_B = \frac{3.3V - 0.7}{R_B} = \frac{3.3V - 0.7}{220\Omega} = 0.011 \text{ A} \tag{1}$$

Moreover, the software was designed for data acquisition and visualization by the E-nose, which runs in a PC that sends and receives Arduino card information by serial communication.

### 2.3.2. Measurement Protocol

Figure 4 illustrates the E-nose scheme proposed for the response measuring and visualization of every gas sensor selected for detection of VOC's emitted by the skin, as possible stress indicators [44].

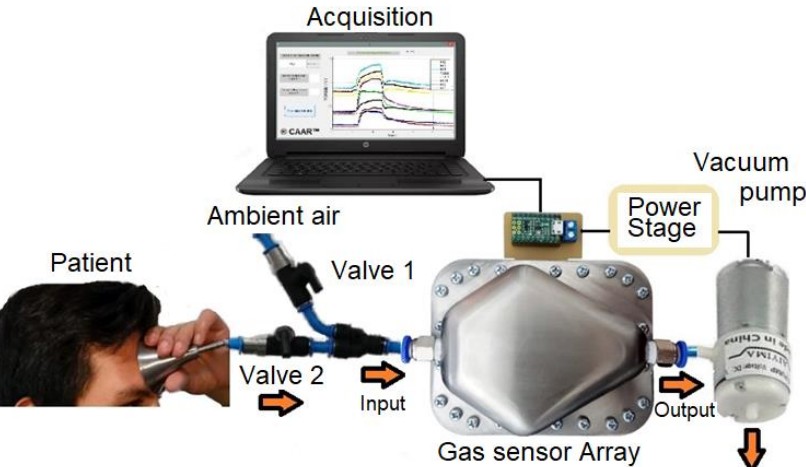

**Figure 4.** Measurement scheme of Volatile Organic Compounds (VOCs) emitted by the skin through the E-nose.

For getting a better VOC's concentration response, a metal funnel was located on the participant's forehead for 5 min, doing a pressure to avoid losing compounds on the outside. After 3 min passed since the funnel was located on the participant, a vacuum pump is activated for purging the measurement chamber, allowing the air passing of the environment. In this way, the heat dispersed by sensors is extracted, which could affect the measurement result. In this stage, valve 2 is closed and valve 1 is opened to purge the piping circuit. After 2 min of purging, valve 1 is closed for allowing the VOC's to pass through the measuring camera. Furthermore, the analogic Arduino outputs gather and send the information of every sensor via serial communication made by the PC, in which the skin sensor's behavior can be observed in real-time.

### 2.4. Electromyography System

The physiological variations on the muscle fiber membrane generate myoelectric signals that can be measured. Therefore, the technique used for measuring and analyzing these signals is known as electromyography. It is commonly performed through Ag-AgC1 electrodes that convert the muscle's ionic current into an electric current [41]. Generally, the amplitude voltage is within the range of $\pm 5$ mV and the frequency content ranges are from 6 Hz to 600 Hz, with a frequency dominant range from 20 Hz to 150 Hz [60].

#### 2.4.1. Design

For this research, an instrumentation amplifier (AD620A) (Analog Devices, Wilmington, MA, USA) from the "Analog Devices" maker was used. This amplifier has a function called Common-Mode Rejection Ratio (CMRR) of high state from 120 dB to 130 dB with gain characteristics from 10 to 1000 and high impedance input: 10 GΩ, which can be applied in data gathering devices [61]. Equation (2) describes the gain calculation ($G$):

$$G = \frac{49.4\,\text{k}\Omega}{R_G} + 1 \tag{2}$$

where setting the $R_G = 100\ \Omega$, an approximate gain equal to 495 was obtained. Furthermore, a filter stage was added for attenuating the noise induced by electromagnetic sources, movement, and even the participant's breathing [61]. Finally, the amplification stage was performed with a variable gain for the measure's taking.

The acquisition measurement scheme can be observed in Figure 5. A Raspberry pi 3B+ card was used to control and acquire the EMG signals, coupling an ADS1015 module (Texas Instruments, Dallas, TX, USA) made by the "Texas Instruments" company. It has an Inter-Integrated Circuit (I2C) communication protocol, four analog inputs, and 3.3 kHz sample rates.

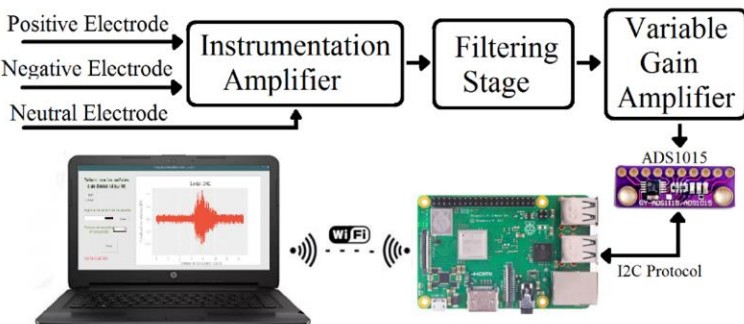

**Figure 5.** Surface electromyogram (EMG) measurement scheme.

For the EMG device, a filtering stage was used. It consists of two "Butterworth" type filters of the second order, of −40 dB/decade. In consequence, when the set cutoff frequency is overpassed 10 times, the filter output will be −40 dB concerning the input. Figure 6 shows the schematic circuit to eliminate the noise produced by the movement and the participant's breathing. A 30 Hz High-Pass Filter (HPF) in a voltage source configuration controlled by voltage and a Low-Pass Filter with a 416 Hz set cutoff frequency were implemented for the EMG system's development.

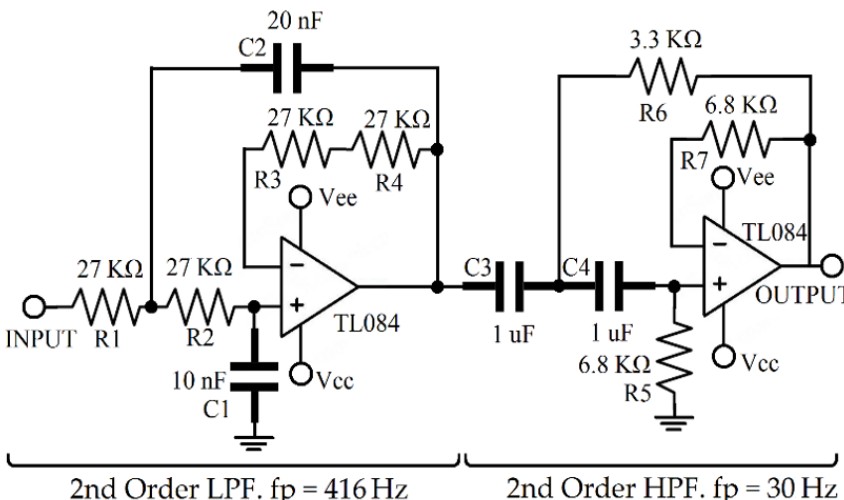

**Figure 6.** Electric circuit for the filtering stage.

The procedure for estimating each of the resistance values and filter circuit capacitors is described below.

Initially, a 416 Hz cutoff LPF frequency is established and an adequate value (preferable commercial) is chosen for C1 between 100 pF and 0.1 uF, so that C1 = 10 nF, the value of C2 = 2 × C1 value is set; finally, the value between R1 and R2 is given by Equation (3), where *Wc* is the cutoff frequency in radians/second. In this stage, the values between R3 and R4 must make the double of R1 [62]:

$$R = \frac{0.707}{W_c \times C} = 27\,\text{k}\Omega \tag{3}$$

Owing to the movement or sudden vibrations in the EMG system, participants' breathing can induce a Direct Current (DC) compound in the EMG gathered signal [61]; for suppressing it, the 30 Hz cutoff HPF frequency was set, calculating the filter parameters in this way: a $C3 = C4 = 1$ uF value is chosen. In this case, the $R7$ value is given by the Equation (4), where $R6 = R7 \div 2$ and the commercial closest resistance was 3.3 k$\Omega$. Finally, for minimizing the deflected DC, the condition $R5 = R7$ is set [62].

$$R7 = \frac{1.414}{W_c \times C} = 6.8 \text{ k}\Omega \tag{4}$$

2.4.2. Measurement Protocol

The electrode setting for EMG surface measuring was made by following the guidelines in the SENIAM regulation for the EMG measuring in upper trapezius muscle [63], as is shown in Figure 7. The electrodes were obtained from the "LifeCare" company, composed of an Ag/AgCI electrode with solid gel for improving conduction, snap connection where these electrodes are hypoallergenic.

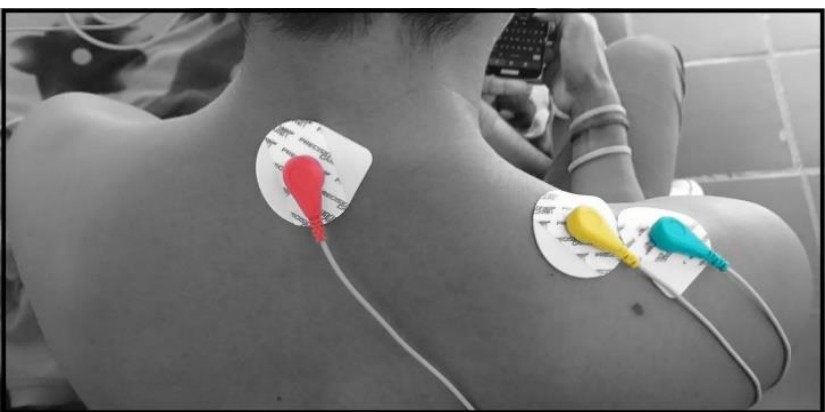

**Figure 7.** Electrodes location for EMG testing.

The data acquired was taken through a graphic interface over the "Raspbian" environment with Python language using the Tkinter library (see Figure 5). In total, two measures in each session were acquired, from stress and relaxation time for 15 s with a 1000 samples per second sample rate. Likewise, a sample was acquired during the execution of a voluntary movement (i.e., in the shoulder) in every student, for then establishing a measurements group and data sorting. It is important to clarify that the EMG system tests were not performed on female students since they expressed it would be uncomfortable and risky during the COVID-19 pandemic. Thus, the above could take on uncertainty about the acquired measures by the other systems [64,65].

*2.5. Heart Rate Variability*

For the physiological signals' gathering, two channels were used, locating the clamp electrode, as is shown in Figure 8. For the signal ECG gathering, the ADS1298ECG-FE module (Texas Instruments, Dallas, TX, USA) from "Texas Instruments" company was set. This module has 8 analog input channels, and it allows measuring the ECG signal in 12 derivations with a 24 bits resolution.

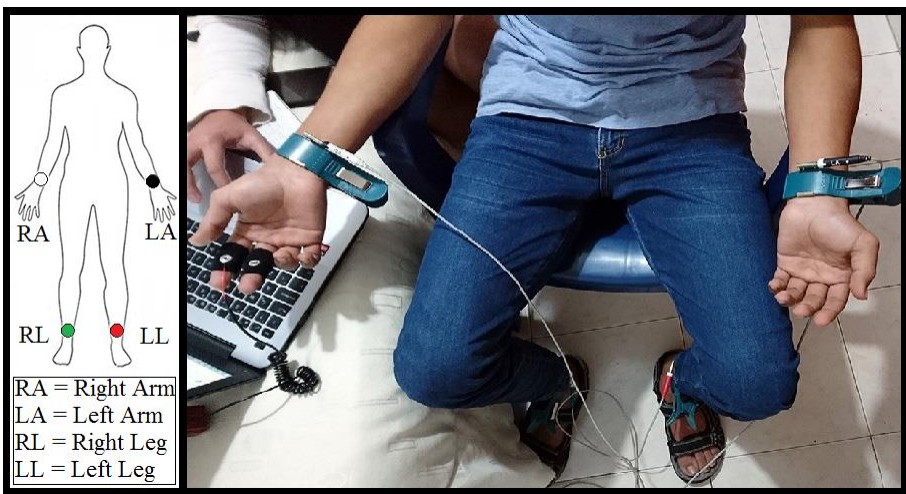

**Figure 8.** Electrode location for ECG signal.

For the R points' location and setting in the ECG gathered signals, an algorithm was designed in Python, leaning on the derivative (see Equation (5)) of the ECG signal, which allows finding the precise localization of each R peak, giving robustness when compounding the HRV signal. Once all R peaks are detected and the knowledge of sample rate with which ECG was acquired, the time between every R peak can be determined; moreover, the concatenation between R peaks represents the HRV signal.

$$y[n] = x[n] - x[n - 1] \qquad (5)$$

*2.6. Processing Methods*

The following methods were used for data analysis.

### 2.6.1. Linear Discriminant Analysis (LDA)

The linear discriminant analysis (LDA) is a data analysis method, it was proposed for binary class problems in 1936 by Fisher [66]. Its goal is to find the data projection that minimizes the variance and maximizes the distance between each class of each measurement that compounds the dataset, guaranteeing in that way the maximum separability between classes, projecting the original matrix data in an inferior dimensional space similar to the PCA (Principal Compounds Analysis). This technique can be used in reduction dimensionality problems such as the previous step for the pattern's sorting and automatic learning [67]. The algorithm calculates the separability between different classes (this is known as between-class variance), next, it determines the distance between the mean and each class sample, known as within-class variance or within-class matrix. Finally, the algorithm constructs an inferior dimensional space that maximizes the variation within the class [68].

### 2.6.2. K Nearest Neighbors (KNN)

The K nearest neighbors (KNN) is an automatic learning method, based on a distance function that determines the similarity or difference between two distances. Commonly, the Euclidean distance is used (see Equation (6)), each distance $x$ is represented by an attribute vector $\langle a_1(x), a_2(x), \ldots, a_n(x) \rangle$, where $a_i(x)$ describes the *i*-th $x$ attribute value [69]:

$$d(x, y) = \sqrt{\sum_{i=1}^{n} (a_i(x) - a_i(y))^2} \qquad (6)$$

The performance of the algorithm depends mainly on the distance metric used for the identification of the nearest neighbors and the number of set neighbors, it shows better results when it is applied on large datasets and with reduced dimensions [70].

### 2.6.3. SVM

Support Vector Machine (SVM) was developed in 1995 by Cortes and Vapnik for binary sorting [71], it is focused on the class separation and ideals separation hyperplane that maximizes the margin between the closest points within the classes [72]. When a linear separator cannot be found, the dataset becomes separable linearly (this projection is made through kernel techniques). Like other supervised learning algorithms, the SVM is trained with a labeled dataset that gives a learning base for the future data sorting, assigning them to a group or another one that is separated [73].

### 2.7. SISCO Inventory

The SISCO inventory is a psychometric instrument that allows measuring the stress level in university students. It can be self-taken, and it can be responded to in an individual or group setting. It is structured in 37 items that allow: Identifying if the survey respondent is an adequate candidate or not by answering the inventory (he can be the right candidate if during the semester he has had worriment or nervousness, if not, he cannot be a candidate), determining the academic stress level intensity, and identifying the environmental demands that are considered stressing stimulus by the survey respondent [74]. Its application was made virtually by qualified staff from the University of Pamplona from the Psychology program. Furthermore, every participant gave their permission for the use of the questionnaires through an informed consent which was shown to them before starting the academic stress inventory application, SISCO.

## 3. Results

### 3.1. GSR Responses

Immediately after the location of the electrodes on the fingers, the signal registered by the sensor response was started, and the sensor response was measured in volts. Figure 9 shows the response acquired during the relaxation state. Therefore, the initial measured value of the skin electric characteristic of each person is different for its physical features, however, it was observed that during the test, the resultant wave-shape behavior always tends to be logarithmic when the participant is in a relaxation state.

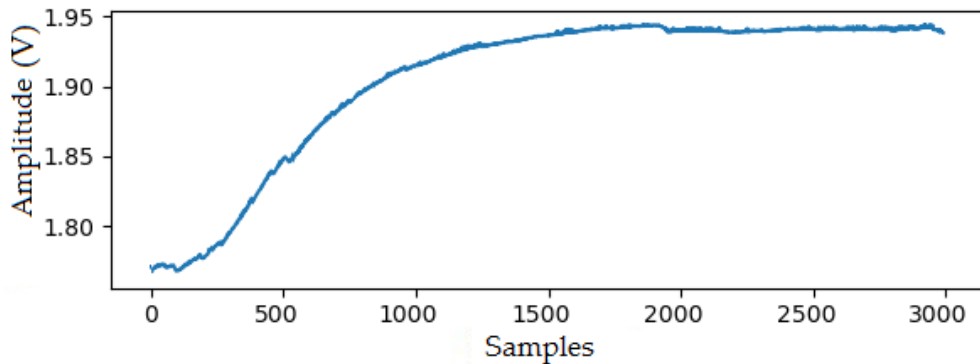

**Figure 9.** Galvanic Skin Response (GSR) signal from a participant in a relaxed state.

Figure 10 shows the GSR signal plot acquired from the same participant during the measuring, corresponding to the stress state where non-random amplitude variations were observed, and an important difference in comparison to the wave shape obtained in the relaxation state measurement that allows discriminating visually between both states.

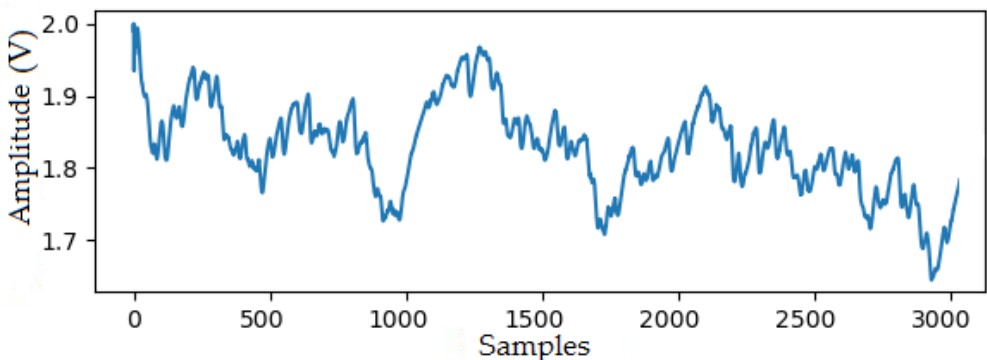

**Figure 10.** GSR signal from a participant in a stressed state.

In an empirical way, it was set at a 5 min minimum period for the GSR signal measuring, since the measurements taken during shorter periods can generate wrong interpretations.

GSR Data Processing

After the acquisition of GSR signals, these were organized in an array where every signal was located in the rows and every signal´s data in the columns. This dataset was normalized using the "StandardScaler" function from the "ScikitLearn" library in Python programming language. This function can obtain the mean and scale the variance in a unitary way. In this stage, the LDA algorithm from the same library was applied, where the resulting LDA algorithm factors were used as training and validation by using the cross-validation method "k-folds", with k = 5 for the Support Vector Machine (SVM) algorithm with the parameters correspondent to the linear kernel.

Figure 11 illustrates the graphic representation of the SVM algorithm response, and the distance to the hyperplane according to the samples analyzed. Through this method, a 100% success rate of classification with the GSR signal was obtained.

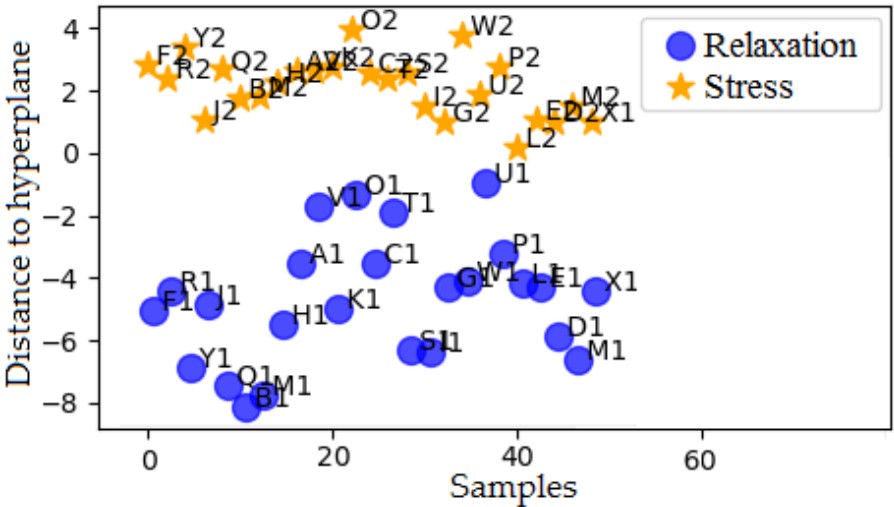

**Figure 11.** The graphical response of the Support Vector Machine (SVM) algorithm in the GSR dataset.

*3.2. E-Nose Responses*

From each measure taken by the E-nose, 8 signals corresponding to each gas sensor that comprises the sensor array were stored. Each sensor response was measured in volts: the lineal base was eliminated in every signal and they were stored concatenating the 8 signals in a single vector; in that order, every sensor was labeled. Thus, the vectors corresponding to every participant's measurements were stored in a matrix, and afterward, they were standardized using the "StandardScaler" instruction. It is important to mention

that standardizing is a common requirement for many automatic learning algorithms. The "ScikitLearn" library was applied to the standardized dataset. Figure 12a depicts the response in Two-Dimensional (2D), where the orange color represents the samples corresponding to the measurement in a stress state, and in blue, the samples acquired in a relaxation state.

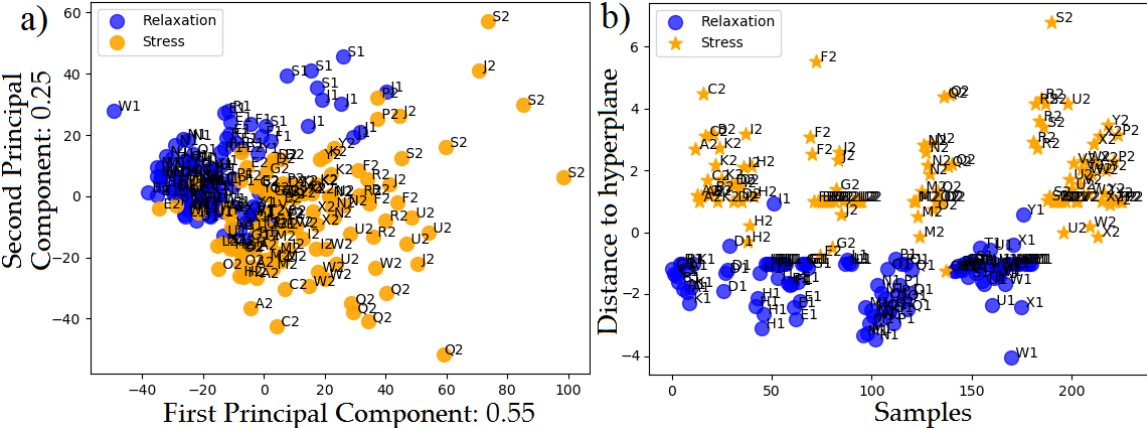

**Figure 12.** The graphical response of the E-nose classification algorithms. (**a**) Response in Two-Dimensional (2D) of samples corresponding to the measurement in a stress state, and in a re-laxation state; (**b**) Graphic of each sample according to the hyperplane distance.

The two first compounds resulting from the dimensionality reduction made by the PCA over the dataset and which represents the 80% variance of data were used for training and validation, applying the cross-validation method "k-folds", with k = 5 from the k nearest neighbors algorithm (KNN). Furthermore, they were established as an input argument: $n$-neighbors = 5, metric = 'minkowski', and $p$ = 2, where $n$-neighbors corresponds to the number of neighbors that are going to use the algorithm for data aggrupation. By using the Euclidean distance, 88.9% precision was attained in the measures sorting. On the other hand, another additional result was obtained with the PCA which was used for the input of SVM algorithm training and testing by the cross-validation method "k-folds", with k = 5. The linear kernel was set and 90% was obtained in the measurements classification, in Figure 12b is shown the graphic of each sample according to the hyperplane distance.

### 3.3. EMG Responses

Figure 13 depicts an upper trapezius muscles EMG signal when a voluntary shoulder movement is done. In Figure 13a, the amplified signal is shown without filtering, and in Figure 13b, the filtered signal, the DC compound has been suppressed and the measure´s variable is given in DC volts, where the signal was acquired during 10 s with a 1000 samples/s sampling rate.

The measurements acquired during a voluntary movement were also used in the feature extraction for the sorting training algorithm. By including the voluntary movement characteristics, we can validate that during the stress and relaxation state, the EMG device was not altered voluntarily. The extracted features for these three labels were: mean absolute value (MAV), waveform length (WL), zero crossings (ZC), slope sign changes (SSC), the variance of EMG (VAR), log detector (LD), difference absolute mean value (DAMV), difference absolute standard deviation value (DASDV), difference variance value (DVARV), and average amplitude change (AAC), described in detail in Reference [75]. All these features were concatenated in a vector for each sample, all taken measures' features were stored in a matrix and they were normalized with the "StandardScaler" function. Afterward, the LDA algorithm was implemented for the processing of the dataset.

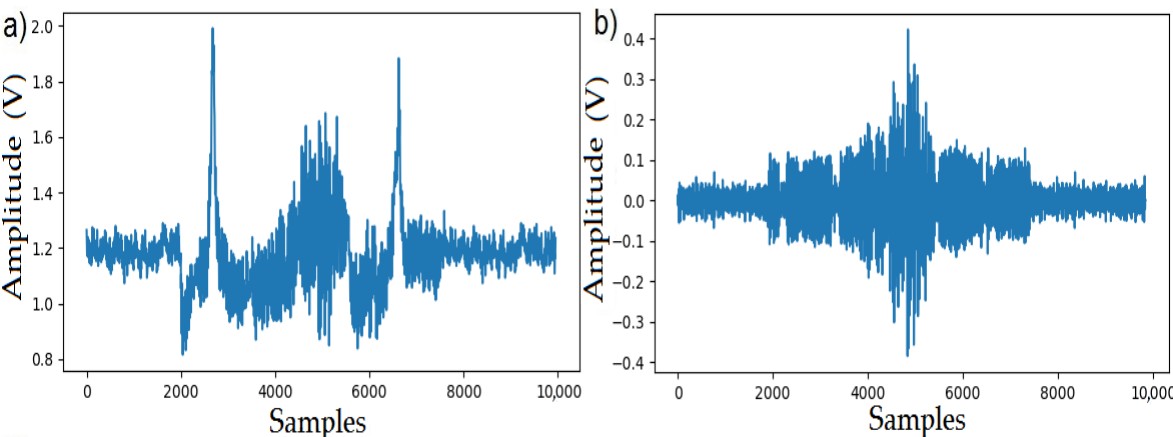

**Figure 13.** EMG signal: (**a**) Unfiltered signal, (**b**) filtered signal.

Figure 14 illustrates the algorithm response where the resulting factors were crossed with the SVM algorithm with lineal kernel and using the cross-validation method "k-folds", with k = 5, achieving a 90% success rate of classification over the EMG features set.

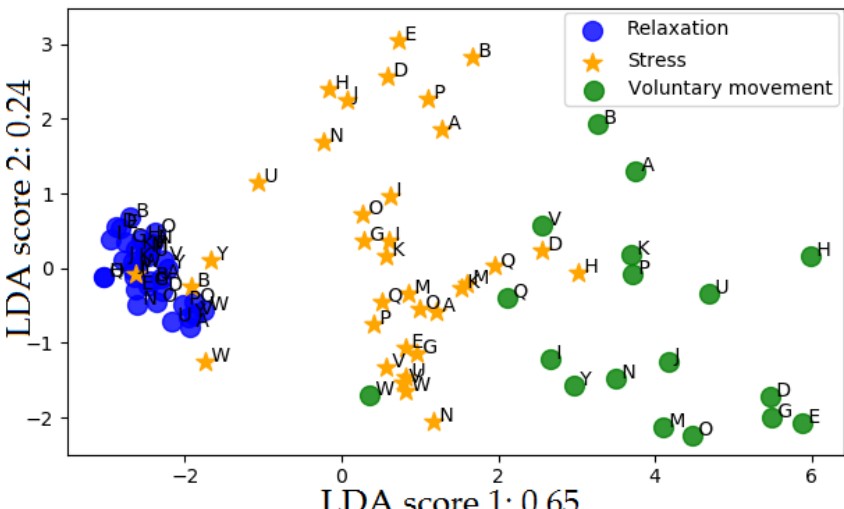

**Figure 14.** The graphical response of the linear discriminant analysis (LDA) algorithm on the EMG feature set.

Regarding EMG signal, the scatter plot (see Figure 14) shows that the classes that make up the dataset of the acquired measures (i.e., relaxation, voluntary movement, and stress) are significantly separable, this indicates that the characteristics extracted from each signal in the time domain provide relevant information for pattern recognition of EMG signals. Besides, different investigations have used this type of time-domain characteristics for the identification and classification of muscle movements in different scenarios of daily life [15,76,77]. For applying the LDA and SVM, the dataset was normalized previously with the "StandardScaler" function.

*3.4. HRV Responses*

The ECG module contains a programmable gain amplifier and a sample rate of 500 samples/s, allowing the ECG signal acquisition in an optimal resolution, as is shown clearly in Figure 15. Besides, in the data gathering and visualization software, digital filters with a variable cutoff frequency are included, with control and monitoring in real-time of the resulting signal varying the filter parameters.

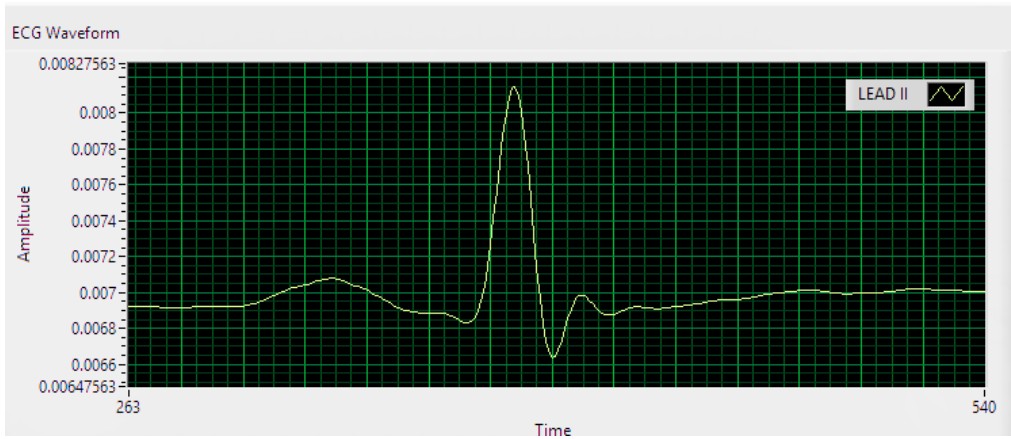

**Figure 15.** ECG signal acquired.

Figure 16 shows the resulting signals in each stage of the developed algorithm, where Figure 16a is the ECG signal, previously filtered, Figure 16b is the derivative of the ECG signal, Figure 16c is the ECG signal's sample with the R peaks detected by the algorithm marked with an asterisk, and Figure 16d is the HRV signal sample extracted from the ECG signal, where amplitude represents the time between R peaks and the horizontal shaft shows the extracted R points.

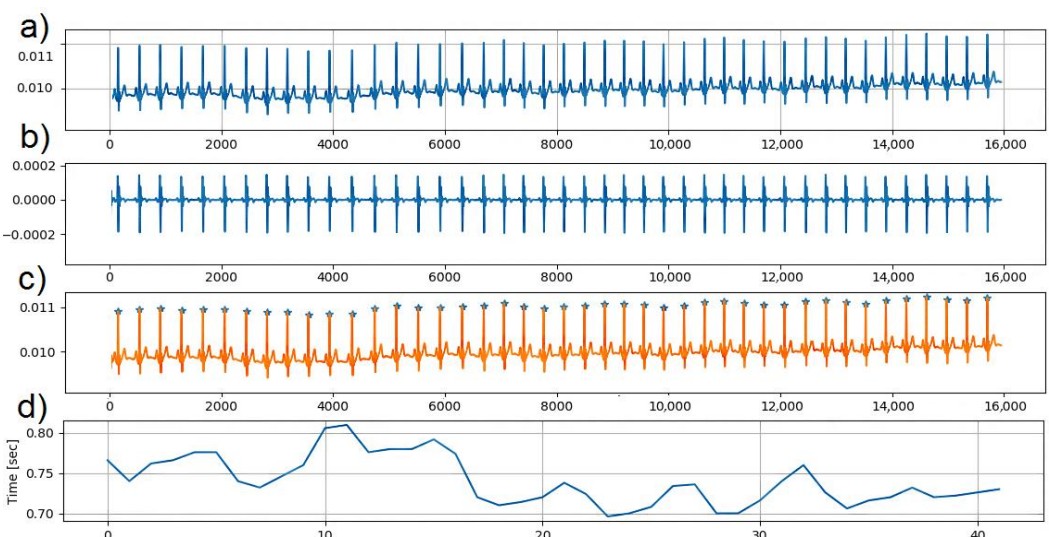

**Figure 16.** Heart rate variability (HRV) signal extraction. (**a**) ECG signal previously filtered (**b**) derivative of the ECG signal (**c**) ECG signal's sample (**d**) HRV signal sample extracted.

After acquiring all ECG measurements during two minutes with a sample rate of 500 samples/s, the algorithm developed in Python was implemented for the HRV signal extraction. Figure 17a shows the ECG acquired signal in relaxation state with all R points identified; in Figure 17b, the HRV signal extracted from 17a is shown, where the amplitude represents the time between R intervals, and it is measured in seconds. On the other hand, Figure 17c illustrates the ECG acquired signal from the same participant in the measure taken in the stress state, and Figure 17d shows the HRV signal extracted from 17c.

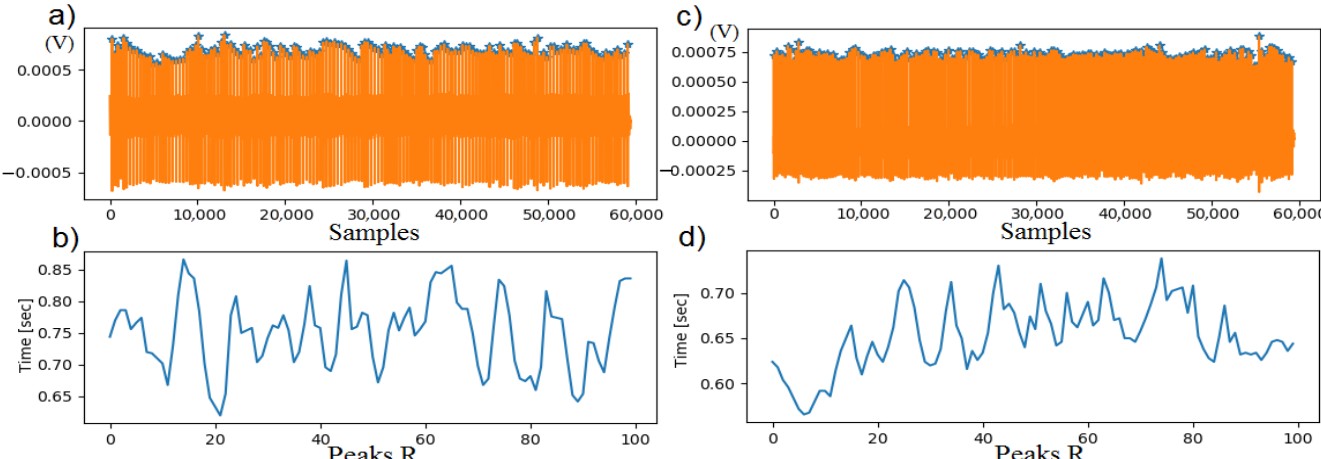

**Figure 17.** ECG and HRV signals: (**a**) ECG in relaxation state, (**b**) HRV signal from (a), (**c**) ECG in a stress state, (**d**) HRV signal from (**c**).

The vectors that contain the HRV signal were comprised by the RR intervals of the ECG signal in the time domain, which were stored in a matrix in which the HRV signals are in the rows and every amplitude signal is in the columns. For using the SVM algorithm, the LDA factors were used as training and validation by using the cross-validation method "k-folds", with k = 5 for the Support Vector Machines (SVM) algorithm. Finally, 88% of classification was obtained, and the algorithm's response is shown in Figure 18.

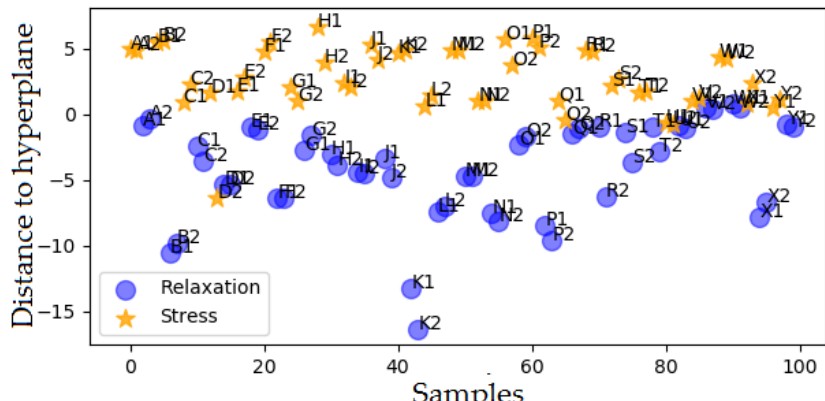

**Figure 18.** The graphical response of the HRV signal classification algorithm.

*3.5. SISCO Method Analysis*

Through SISCO inventory, it is possible to distinguish three stress levels, these levels are determined through the intensity with which a situation is considered as stressful from the student's perspective. According to the score obtained after the test performance, these levels can be sorted as follows: 0 to 33 indicates a mild stress level, 34 to 66 represents a moderate stress level, and scores between 67 to 100 are considered a deep academic stress level.

Consequently, according to the global psychometric data obtained in the psychological analysis, in the group of student survey respondents, 4% show mild stress levels, 64% are at a moderate stress level, and the remaining 32% show intense stress levels, as represented in Figure 19.

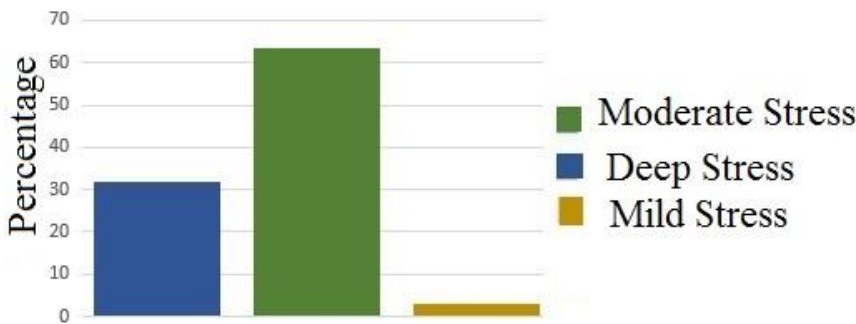

**Figure 19.** Level of stress in the study population.

The following information was obtained among the most representative data: mean equal to 55.44, which keeps within the range equivalent to moderate stress, a 54 grades median, and a 45 points mode. Thus, the standard deviation was equalized to 12.8, and the variance coefficient was 23.

Through the questionnaire, a 10-item "Likert-scale" was applied to identify the environmental demands which are perceived as stressful stimulus. In this section, the results are shown in the relevant order: Teacher´s exams, homework overload, and fear of getting the answers wrong, which were the most representative stressful factors in the students' academic performance.

Besides, in the 18-item section from the SISCO inventory, the symptoms or reactions toward the stressful stimulus are indicated, among which are sleep disorders, anxiety, concentration problems, and headaches. These were the symptoms shown with more frequency among the students. The confrontation strategies were assessed in an 8-item section, among which stood out: Defending their ideas without damaging others, a plan elaboration, homework execution, and information searching about the situation. The professional assistance searching was one of the strategies with which the participating students least related to.

On the other hand, Table 3 illustrates the results of the SISCO inventory and the GSR signals which depicted that there was a correlation between the different stress levels (i.e., Deep, Moderate, and Mild), where the maximum and minimum values from the GSR signal were extracted to determine the sensitivity of the device. It should be clarified, once the analysis was performed with four devices used in this study, the GSR device was chosen for the correlation of the data since it revealed a better classification and correlation of the results. Therefore, it can be seen that there is a correlation both in the SISCO data and the signal measured by the electronic device. It can be seen that the results of the three different stress levels obtained by SISCO can be almost completely matched with the sensitivities calculated from the GSR signals. To perform the comparison of the results, the sensitivity of each of the electrode signals was determined. Therefore, the range taken for this study was 100%, which was related to the highest value obtained in the sensitivities, which was 0.5. Thus, the range of 0–0.165 was established, which corresponds to the mild level of stress, from 0.17–0.33 to moderate stress, and finally, for a deep level of stress, which was 0.335–0.5.

Therefore, among the SISCO tests and the response of the GSR signals, it was possible to obtain a 92% success rate since only two signals (C and X) were not matched correctly.

**Table 3.** Comparison among SISCO inventory and Galvanic Skin Response (GSR) signal.

| Student | SISCO | | GSR | |
|---|---|---|---|---|
| | Score | Stress Level | Sensitivity | Stress Level |
| A | 46 | Moderate | 0.23560 | Moderate |
| B | 70 | Deep | 0.36670 | Deep |
| C | 66 | Deep | 0.32986 | Moderate |
| D | 46 | Moderate | 0.21543 | Moderate |
| E | 44 | Moderate | 0.22256 | Moderate |
| F | 63 | Moderate | 0.32087 | Moderate |
| G | 35 | Moderate | 0.28092 | Moderate |
| H | 74 | Deep | 0.45831 | Deep |
| I | 51 | Moderate | 0.27593 | Moderate |
| J | 66 | Deep | 0.34669 | Deep |
| K | 45 | Moderate | 0.25159 | Moderate |
| L | 71 | Deep | 0.37501 | Deep |
| M | 31 | Mild | 0.12581 | Mild |
| N | 63 | Moderate | 0.23650 | Moderate |
| O | 44 | Moderate | 0.19235 | Moderate |
| P | 61 | Moderate | 0.21188 | Moderate |
| Q | 73 | Deep | 0.34540 | Deep |
| R | 72 | Deep | 0.36484 | Deep |
| S | 54 | Moderate | 0.30226 | Moderate |
| T | 52 | Moderate | 0.26412 | Moderate |
| U | 41 | Moderate | 0.29144 | Moderate |
| V | 49 | Moderate | 0.26504 | Moderate |
| W | 68 | Deep | 0.49330 | Deep |
| X | 57 | Moderate | 0.33670 | Deep |
| Y | 45 | Moderate | 0.24072 | Moderate |

## 4. Discussion

Significant variations were found in the measurements taken during an exam performance in comparison to the other measurements taken in relaxation state through the EMG signal features. The above was because the EMG measurement response changes from one person to another due to their physical characteristics such as body fat percentage, the age, the kind of sports activities done by the students, or the lack of activities that include physical effort. Therefore, it is necessary to count on an efficient normalization method that allows reducing the variability in the EMG signal features, and in this way, responses could be improved through the classification algorithms.

Besides, it can be mentioned that different factors may occur in the GSR signal response, such as a participant's deep breathing execution while the signal is being taken, which can induce an amplitude decrease during the measuring time. For that reason, every student was told to abstain from doing deep breathing during the 5 min measuring time. However, in these cases, the sample was dismissed and was taken again.

In the same way, it was demonstrated that a slight decrease in the HRV response amplitude was acquired in the stress state measures in comparison to relaxation states per student. The above is in line with the hypothesis in which the stressing events induce an increase in the heart rate [16]. However, for future research, it would be important to consider the use of adhesive electrodes that allow a better conductivity, since some students have shown ECG signals with very low intensity or amplitude, which is why it was necessary to modify the algorithm for keeping the R peaks right and recognizing the ECG signal.

Concerning the E-nose response, the commercial sensors selected were capable of detecting the Volatile Organic Compounds emitted by skin, since it was possible to make a differentiation in various categories. We can highlight that these kind of sensors are also sensible to VOC's presence in breath [78]. That is why it is important to handle the E-nose system carefully for avoiding the VOC's emitted by breathing becoming absorbed in the

measuring time, inducing possible confounding factors in the acquired data. The low cost of commercial gas sensors based on metal oxide semiconductors are a good option for their use in multisensorial systems with biomedical applications, in addition to their diversified use in industrial applications [79,80].

It is worth considering that the psychosocial and physical aspects of daily life, such as family, economic, and behavior problems, as well as events that happen in our environment and that we cannot control, for example, in the job, such as an overload of work or being fired, can generate alterations in the student´s mental state. Moreover, other factors such as personal problems with teachers, low academic performance, the loss of a loved one, domestic violence, as well as psychoactive substances consumption, medicines, and unbalanced feeding, can alter the body's response toward different environmental stimulus [81,82]. Experiences like these increase the importance of knowing the current profile and situations that the participant volunteers have been through in this research since it is possible to minimize wrong judgments that can hamper the research advancements and the evaluation of new technologies.

It is important to clarify that the LDA pattern recognition method was used with two main factors (F1 and F2) to apply the SVM algorithm. Therefore, the success rate in data classification was good because the best information was obtained from the dataset before applying SVM. This study was conducted to detect the level of stress during the COVID-19 pandemic as a pilot study carried out on engineering students at the University of Pamplona, where it was not easy to acquire the samples due to the risks caused by COVID-19 at the time of acquiring the samples, and the lack of cooperation on the part of the students. However, some students agreed to perform the different tests with the electronic devices during the virtual exam. It is noteworthy that one of the key points for the success in the acquisition and classification of the samples was because of the fact that the students had two quite noticeable options, one was to think about being able to pass the exam and the other to lose it. Thus, these two scenarios generated in them quite noticeable stress in either of the two situations during the exam.

Consequently, the present study generated interesting results through the detection of stress in academic contexts. Therefore, we can say that this study is the first to be carried out on this subject.

Indeed, there are many articles on the topic of stress using electronic devices and where good results have been obtained in the classification of the measures. For instance, one study investigated the efficacy of the data fusion from off-the-shelf sensors to accurately determine stress in humans, in this case, the SVM reached 100% [83]. In another work, SVM was applied for mental stress detection in University Students. The algorithm obtained 100% specificity to classify the dataset [84].

It is necessary to highlight that no work was found in relation to the combination of LDA + SVM for stress classification.

In this study, we want to highlight the Quintero-Posada and Chon article as they have done a study with different alternative techniques, like spectral analysis, which have emerged as potential tools for the analysis of the electrodermal activity (EDA). These new methods and tools may help us to generate new applications in the future by using the signals information obtained from the electronic devices to monitor the mood or stress of a person for short- and long-duration data records [85].

## 5. Conclusions

Through the methodology proposed by using electronic devices, it was possible to get a high precision in the acquired data corresponding to every state (stress and relaxation state), because these were used in a real situation in which the student was taking a virtual exam. According to the SISCO inventory results of the academic stress, the exams are considered a stressing agent with the most relevance in the academic population.

With the GSR device, a better response could be obtained for detecting stress. Therefore, a 100% success rate was obtained in data classification, and moreover, we can mention

that this device is still one of the most efficient methods for stress detection. This system also allowed recognizing, visually from the gathered wave-shape, the participant's state (stressed or relaxed). However, it is important to keep on classifying the GSR response wave-shape in different situations and environments, since some researchers suggest that the state of mind can alter the GSR signal amplitude [82]. Consequently, it would be important that in future research, each person's characteristics can be defined before its use.

With the VOC's detection technology from the E-nose, 90% classification was obtained; besides, this could be incremented with the application of more advanced pattern recognition techniques. However, though good results were obtained, and it has been the first research conducted for measuring stress in a pandemic situation, it would be worthwhile to compare the proposed system functioning with classical gas analysis techniques such as gas chromatography and mass spectrometry (GC-MS), for being able to validate the proposed protocol adequately.

In this research, only the HRV signal response was assessed with the single-channel device designed, so, for future investigations, a two-channel device could be implemented that allows acquiring the EMG response of the two shoulders to obtain more information that could increase to a 90% success rate and continue exploring a better measurement protocol, since the resulting signal amplitude could be affected for the electrode responses.

Finally, from the HRV signal response, 88% of data classification was achieved, confirming that ECG systems are still a good option in psychophysiological research that expects to measure the person's physiological behavior.

We want to mention that the most significant aim of the study was to try to detect the academic stress during the COVID-19 pandemic by using different kinds of electronic devices, as we wanted to see how the virtual exams could generate stress in the university students and which of those devices could be more efficient to detect it. Thus, it is very important to comment that there were a lot of issues with regards to the sample collections because the students were quite scared, and they did not have much time to collaborate with this study because they had exams. However, at the end of this study, the students agreed and they were willing to participate in the different experiments to obtain results.

Nonetheless, despite the limited number of samples acquired due to the pandemic period and to the difficulty of being in direct contact with the student, we still obtained promising results for further investigations. Therefore, from this first research performed with each electronic device exposed in this article, we hope that we can take larger measurements set after the pandemic, considering more deeply the participants' characteristics and states before performing the test with the sensorial devices and psychological analysis.

**Author Contributions:** Conceptualization, C.M.D.-A. and J.K.C.-G.; methodology, C.M.D.-A., J.K.C.-G. and C.A.A.-R.; software, J.K.C.-G.; validation, J.K.C.-G. and C.A.A.-R.; formal analysis, C.A.A.-R.; investigation, C.M.D.-A.; data curation, J.K.C.-G.; writing—original draft preparation, C.A.A.-R.; writing—review and editing, C.M.D.-A., J.K.C.-G. and C.A.A.-R.; supervision, C.A.A.-R.; project administration, C.M.D.-A.; funding acquisition, C.M.D.-A. All authors have read and agreed to the published version of the manuscript.

**Funding:** A part of this research was funded by Minciencias, project code: 112184468047, grant number 844 (2019), and another part was funded by the University of Pamplona and GISM group.

**Informed Consent Statement:** Informed consent was obtained from all subjects involved in the study.

**Data Availability Statement:** The data presented in this study are available on request from the corresponding author.

**Acknowledgments:** The authors of the manuscript would like thank Sandra Padilla and his student Jhon Tierradentro from the psychology program of the University of Pamplona for their participation in the use of the SISCO tool.

**Conflicts of Interest:** The authors declare no conflict of interest.

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
