# Peer review of "Electronic Devices for Stress Detection in Academic Contexts during Confinement Because of the COVID-19 Pandemic"

_electronics, doi:10.3390/electronics10030301_

Round 1

Reviewer 1 Report

This manuscript presented diverse methods to measure the stress in academic contexts because of the COVID-19 pandemic. It is an interesting work for showing a correlation between the COVID-19 pandemic and stress in academic contexts. However, the overall composition and quality of the manuscript should be improved.

Question 1: Please provide data indicating the correlation between individual survey results and actual signal measurement result.

Question 2: The size, coordinate axis, and font size of most figures in this paper should be standardized and unified. Also, the descriptions of each figure should be provided accurately separated.

Question 3: The authors should rewrite the materials and methods. Current materials and methods seem mixed with results.

Question 4: The authors should provide in their conclusion the significance of the study with the academic stress caused by the COVID-19 pandemic.

Reviewer 2 Report

I read with great interest this manuscript. With the methodology proposed by using electronic devices authors got a high precision in discriminating between stress and relaxation states.

I have some comments:

  • Please add more details about subjects characteristics, possibly in a table (i.e. age, sex, race). 
  • Please specify inclusion/exclusion criteria for your study population
  • Please state whether you estimated sample size population.
  • Please correct some typing errors thoroughout the paper
  • Authors did not fill the "Institutional Review Board Statement". Please do that. 

Reviewer 3 Report

This is a good attempt to combine EMG, ECG and GSR to discriminate between baseline and stress condition.  However, data analysis technique especially for GSR does not involve the usual decomposition of the signal into skin conductance level and skin conductance resistance.  It appears that the authors have taken the entire signal and try to discriminate between the two conditions.  It is highly recommended that the authors see the recent review paper (Posada-Quintero and Chon, Sensors) for properly analyzing GSR data and implement it for their analysis. 

Moreover, it is not clear if HRV signal is just based on RR intervals or spectral analysis of RR intervals.  It appears that the authors have taken SVM and LDA directly to RR intervals for both conditions.  Similarly, data analysis involving EMG is rather limited.

it is quite surprising that the results based on SVM and LDA on their acquired signals show such accurate discrimination results between baseline and stress induced by taking exams.  There have been numerous studies examining stress conditions and none of the reports in the literature have been able to provide such accurate discrimination.  Hence, more details need to be provided for the experimental conditions.  Please also provide other background results on this subjects and why the current results provide much more accurate results than other publications.

Round 2

Reviewer 3 Report

note that the analysis approaches presented in the paper by Posada et al. do not require long duration data records; it can be done with the same amount of data used in this work.  Please make this correction to the revised paper.

Author Response

Thank you for your remark, We have already corrected and included your recommendation in the manuscript. We appreciated your support.

In this study, we want to highlight the Quintero-Posada and Chon article as they have done a study with different alternative techniques like spectral analysis which have emerged as potential tools for the analysis of the Electrodermal activity (EDA).

These new methods and tools may help us to perform new applications in the future by using the signals information obtained from the electronic devices to monitor the mood or stress of a person for short and long duration data records [85].

85. Posada-Quintero, H.F.; Chon, K.H. Innovations in Electrodermal Activity Data Collection and Signal Processing: A Systematic Review. Sensors. 2020, 20, 479. 10.3390/s20020479.
